# Magnetic Water Treatment: An Eco-Friendly Irrigation Alternative to Alleviate Salt Stress of Brackish Water in Seed Germination and Early Seedling Growth of Cotton (*Gossypium hirsutum* L.)

**DOI:** 10.3390/plants11111397

**Published:** 2022-05-25

**Authors:** Jihong Zhang, Quanjiu Wang, Kai Wei, Yi Guo, Weiyi Mu, Yan Sun

**Affiliations:** State Key Laboratory of Eco-Hydraulics in Northwest Arid Region of China, Xi’an University of Technology, Xi’an 710048, China; zhangjihong_eric@163.com (J.Z.); 18291869766@163.com (K.W.); 15503635823@163.com (Y.G.); sunyan199058@126.com (Y.S.)

**Keywords:** irrigation water quality, magnetized water, seed germination, cotton seedling, photosynthesis, biomass partition

## Abstract

Magnetized water has been a promising approach to improve crop productivity but the conditions for its effectiveness remain contradictory and inconclusive. The objective of this research was to understand the influences of different magnetized water with varying quality on seed absorption, germination, and early growth of cotton. To this end, a series of experiments involving the seed soaking process, germination test, and pot experiment were carried out to study the effects of different qualities (fresh and brackish water) of magnetized water on seed water absorption, germination, seedling growth, photosynthetic characteristics, and biomass of cotton in 2018. The results showed that the maximum relative water absorption of magnetized fresh and magnetized brackish water relatively increased by 16.76% and 19.75%, respectively, and the magnetic effect time of brackish water was longer than fresh water. The relative promotion effect of magnetized brackish water on cotton seed germination and growth potential was greater than magnetized fresh water. The cotton seeds germination rate under magnetized fresh and magnetized brackish water irrigation relatively increased by 13.14% and 41.86%, respectively, and the relative promoting effect of magnetized brackish water on the vitality indexes and the morphological indexes of cotton seedlings was greater than magnetized fresh water. Unlike non-magnetized water, the net photosynthetic rate (*P_n_*), transpiration rate (*T_r_*), and instantaneous water use efficiency (*iWUE*) of cotton irrigated with magnetized water increased significantly, while the stomatal limit value (*L**_s_*) decreased. The influences of photosynthesis and water use efficiency of cotton under magnetized brackish water were greater than magnetized fresh water. Magnetized fresh water had no significant effect on biomass proportional distribution of cotton but magnetized brackish water irrigation markedly improved the root-to-stem ratio of cotton within a 35.72% range. Therefore, the magnetization of brackish water does improve the growth characteristics of cotton seedlings, and the biological effect of magnetized brackish water is more significant than that of fresh water. It is suggested that magnetized brackish water can be used to irrigate cotton seedlings when freshwater resources are insufficient.

## 1. Introduction

Freshwater scarcity has become a global problem to sustainable agricultural development especially in arid and semi-arid areas [1,2]. The exploitation and utilization of brackish water have become an important way to alleviate the shortage of freshwater resources and guarantee food production [3,4]. However, brackish water irrigation can result in soil quality deterioration, such as salt accumulation, reduced water conductivity, insufficient oxygen content, organic matter deficiency, etc. [5,6]. Such problems trigger cell osmotic stress, physiological drought, ionic toxicity, and nutrient scarcity, resulting in abnormal metabolic function and reduced crop productivity [7]. It should be noted that when the salinity of brackish water surpasses 3 g/L, plant physiological growth exhibits salt stress symptoms [8]. Therefore, it has become essential to continuously seek new technologies to improve irrigation water quality and alleviate the salt stress of brackish water.

Magnetic water treatment, as a potential and eco-friendly new technology [9], is widely used in the field of agriculture [10]. Studies and practices show that water can be magnetized by application of a magnetic field [11,12,13]. Magnetization of water induces beneficial changes to its micro and macro physical and chemical properties. The activity of magnetized water (i.e., the ability of water to interact with other substances, such as solubility, reaction rate, etc.) is obviously enhanced [14,15], which is very significant in improving water availability and crop stress resistance [16,17]. Khoshravesh et al. (2011) [18] and Mostafazadeh-Fard et al. (2010) [19] found that soil water content of magnetic water irrigation increased by 7.5% compared to non-magnetized water, magnetized water had higher soil water content after 1, 2, and 3 days of irrigation. Selim et al. (2019) [20] pointed out that magnetized water enhanced the drought resistance of wheat to a certain extent, and dry weight, total soluble sugar content, total water content, total free amino proline and acids improved by 32%, 17%, 12%, 73%, and 27%, respectively.

Recently, the application of magnetized brackish water has attracted extensive attention as a promising practice for alleviating the toxicity and other negative effects of brackish water and promoting crop productivity. Mohamed (2013) [21] found that the fresh and dry weight of tomato plants irrigated with magnetized brackish water increased compared with the weights of plants treated with non-magnetized water. Surendran et al. (2016) [22] showed that the yield of eggplant irrigated with magnetized brackish water improved by 17.0%. Hozayn et al. (2019) [23] indicated that the biological yield, straw yield, and grain yield of wheat under magnetized brackish water irrigation improved by 26.99%, 33.97%, and 19.24%, respectively. In addition, magnetized water irrigation can also improve soil environment and crop stress resistance.

Furthermore, magnetized water irrigation has been applied in improving crop seed germination and seedling growth [24,25]. Aghamir et al. (2017) [26] pointed out that magnetized water significantly enhanced the salt tolerance seeds and promoted the early growth of maize (*Zea mays*). El Sayed. (2014) [27] reported that magnetized water irrigation observably improved the growth, chemical composition, and yield production of broad bean (*Vicia faba* L.) seedlings. Moreover, the promotion effect of seedling growth by magnetized water irrigation is closely linked to the improvement in crop light energy utilization. Alfaidi et al. (2017) [28] demonstrated that magnetized water significantly increased the contents of chlorophyll and carotenoid in guinea grass leaves. Liu et al. (2019) [29] found that magnetized water improved the net photosynthetic rate and water use efficiency of *Populus ×  euramericana ‘Neva’*, while decreased the transpiration rate and limited stomatal conductance as compared to non-magnetized water irrigation.

To sum up, magnetized water irrigation has been widely studied on food crops and vegetables, while the effect of magnetized water irrigation on fiber crops, especially fiber crops in the seedling stage, is rarely reported. Cotton (*Gossypium hirsutum* L.) is the most important fiber crop and the largest agricultural trade commodity in the world [30,31], which is irrigated primarily in arid and semi-arid areas [32]. As important irrigation water sources, fresh and brackish water have an important impact on the yield and quality of cotton, while the seedling stage is most sensitive to external factors. Therefore, the hypothesis to be studied is whether the quality of magnetized water affects seed water absorption, germination, and seedling growth in different ways. In this study, the different influences of magnetized fresh and brackish water on early growth of cotton were compared and analyzed via seed water absorption characteristics, germination parameters, seedling growth, biomass, photosynthesis parameters, and chlorophyll content. The results can provide a reference for the efficient use of brackish water in arid and semi-arid regions. 

## 2. Materials and Methods

### 2.1. Study Site and Irrigation Water Quality Description

The experiment was carried out in the Bazhou irrigation experimental station in Korla, Xinjiang, China. The station is located in the middle of Eurasia and the alluvial plain of the Konqi River at the edge of the Tarim Basin in the southern foothills of the Tianshan Mountains (41°35′ N, 86°10′ E, and 901 m a.s.l.) [33,34]. The study area is an important cotton growing area in Northwest China, which has a typical inland arid climate. The maximum potential evaporation and annual average precipitation are 2788.2 and 58 mm, respectively. The annual sunshine and frost-free period are 3036.2 h and 192 days, respectively [35]. The average wind speed and maximum wind speed are 2.4 and 22 m s^−1^, respectively, the maximum temperature is 30.9 °C and the minimum temperatures is 15.8 °C [36,37]. The fresh and brackish water used in the study are diluted or proportioned according to the ion composition of local irrigation water, and their chemical composition is shown in Table 1.

### 2.2. Magnetizer and Magnetized Water Device

The CHQ type external permanent magnetizer made by Shanghai Juncai magnetic material Co., Ltd., China, was used in the experiments. It is made of sintered Rufe-B, with a magnetic field strength of 300 mT and magnetic field area of 8 × 10 cm. The magnetized water device consists of a water box, PVC water pipeline, magnetizer, and peristaltic pump. The water pipeline diameter is 2.5 cm and the effective magnetic length is 10 cm. The water box is used to store 100 L of irrigation water, and the peristaltic pump is used to control the water flow rate in the closed circulate pipeline (Figure 1). A part of the circulate pipeline is placed at the two poles of the magnetizer, and the water flow direction is perpendicular to the direction of the magnetic induction line. When making magnetized water, the magnetic time was set to 30 min with a 14.7 L min^−1^ water flow rate.

### 2.3. Experimental Design

The fresh and brackish water were treated by the magnetized water device, and 4 treatments were generated: non-magnetized fresh water (NF), magnetized fresh water (MF), non-magnetized brackish water (NB), and magnetized brackish water (MB). A series of tests consisting of the seed soaking process, seed germination test, and pot experiments were performed with the 4 treatments of water to compare and analyze the effects of different qualities of magnetized water on cotton growth, in 2018.

#### 2.3.1. Experiment 1—Soaking Process

The seed water absorption test was adopted from Shafaei et al. [38] with a slight modification. In total, 100 air-dried cotton seeds were randomly selected and weighed, then placed in glass beakers containing 200 mL of treated water and placed in an incubator at 28 °C. The weight of water absorbed by cotton seeds was measured at intervals of one hour to determine equilibrium water content (the water in the seed body was saturated). After attaining equilibrium water content, the samples were removed from the beakers and placed on hygroscopic papers to eliminate the excess water, and soaked seeds were then weighed. A digital stopwatch (MH2000C, accuracy ± 0.01 g Shanghai Maohong Electronic Technology Co., Ltd., Shanghai, China) was used to record soaking duration and an electronic balance to measure the weight of the sample before and after immersion. In order to minimize errors, all tests were performed in triplicate. The relative water absorption and water absorption rate of cotton seeds were calculated according to Equations (1) and (2) [39], respectively.
(1)$$W_{r,t}=\frac{W_{a,t}-W_{b}}{W_{b}}\times 100\%$$
(2)$$V_{s,t}=\frac{W_{a,t}-W_{a,t-1}}{\Delta t}$$
where *t* is immersion time (h); *W_r,t_* is relative water absorption at time *t*; *V_s,t_* is water absorption rate at time *t* (g h^−1^); *W_a,t_* and *W_a,t_*_−1_ are weight of seeds after immersion at time *t* and *t*−1, respectively (g); *W_b_* is weight of seeds before immersion (g).

#### 2.3.2. Experiment 2—Germination Tests

The experimental Petri dishes for the seed germination tests were marked with all testing combinations replicated three times [40]. Two sheets of crepe cellulose paper were placed on a Petri dish and moistened with 20 mL of treated water. Fifty cotton seeds of same size were randomly selected and planted on top of the moistened filter paper. After planting, all tests were placed inside an incubator at 28 ± 1 °C under a 12 h photoperiod, and the light intensity was 800 Lx. After 4 d, an additional 5 mL of the corresponding treated water was added to the filter paper to prevent the paper drying. A seed was recorded as germinated when the radicle length from the seedcoat was at least 2 mm. The germination condition was evaluated after 8 d according to Zhang et al. [41]. The germination potential (*GP*), germination rate (*GR*), germination index (*GI*), and vigor index (*VI*) was calculated as follows [42]:(3)$$GP=\frac{N_{4}}{N_{T}}\times 100\%$$
(4)$$GR=\frac{N_{8}}{N_{T}}\times 100\%$$
(5)$$GI=\sum \frac{G_{t}}{D_{t}}$$
(6)VI=GI×L
where, *N*_4_ and *N*_8_ represent the number of germinated seeds at the 4th and 8th days, respectively; *N_T_* represents the total number of seeds; *G_t_* represents the number of germinated seeds at t day (*D_t_*); *L* represents the average seed radicle length at the 8th day.

#### 2.3.3. Experiment 3—Pot Experiments

The temperature and rainfall during field pot experiments in 2018 are shown in Figure 2. Random pot trials were conducted on cotton fields, and the pot experiment was statistically assessed using an analysis of variance. With a total of 12 pots, four treatments were set up based on varied water quality (NF, MF, NB, and MB), and each treatment was performed three times. The air-dried field soil from the upper 20 cm of the cotton field was used, which was sandy loam, according to the USDA’s soil texture classification (64.27% sand, 32.83% silt, 2.90% clay). The total N, available P, available K, ECe (electrical conductivity of soil saturation extract), and pH were 23.81 mg kg^−1^, 15.23 mg kg^−1^, 107.56 mg kg^−1^, 5.47 ds m^−1^, and 7.8, respectively [43]. On 22 April 2018, cotton seeds were sown. Before sowing, 6.8 g carbamide (N 45.4%), 6.4 g compound fertilizer (N 12%, P 18%, K 15%), and 24 g organic fertilizer (organic matter > 35%) was applied per pot. Twenty cotton seeds were sown in each plastic pot (30 cm diameter × 20 cm high), which contained 20 kg of soil mixed with the fertilizers. The irrigation amount per pot was practically the same as the spring irrigation in the field (100 mm), and no longer irrigated at seedling stage. All pots were covered with plastic film mulch to minimize soil evaporation. The emergence rate (*ER*) was counted day by day and seedlings were thinned to four plants per pot after germination (14 d). The removed seedlings were used to measure the seedling activity indexes consisting of seedling height (*Hs*), taproot length (*TLs*), fresh weight (*FWs*), dry weight (*DWs*), and moisture content (*MCs*) [40]. Meanwhile, the retained seedlings were used to measure the morphological indexes including plant height, stem diameter, number of leaves per plant, and area of single leaf every 10 days [33]. The photosynthetic parameters containing the net photosynthetic rate (*P_n_*), stomatal conductance (*G_s_*), intercellular CO_2_ concentration (*C_i_*), and transpiration rate (*T_r_*) were determined by LC Pro SD full-automatic portable photosynthetic instrument (UK ADC), and the stomatal limit (*L_s_*) and instantaneous water use efficiency (*iWUE*) were calculated [44] at 30 d after germination. In the meantime, the SPAD value of chlorophyll was determined by the SPAD-502 Plus Chlorophyll Meter (Konica Minolta China). Finally, the seedlings were removed slowly and the biomass indexes including the dry weight of stem, leaf, root, and total biomass were weighed at 40 d after germination.

### 2.4. Statistical Analysis

All measured data were recorded in Excel 2019 and examined by analysis of variance (ANOVA) using SPSS 22.0 software (IBM Corp., Armonk, NY, USA). Significant differences (*p* < 0.05) between means were identified using the least significant difference (LSD) test. Figures were drawn using Origin 2021 software.

## 3. Results

### 3.1. Water Absorption Characteristics of Cotton Seed

With the increase in soaking time, the relative absorption of magnetized and non-magnetized water by cotton seeds first increased rapidly and then tended to be constant. The maximum relative water absorption (*RWA_m_*) for differently treated water and the time required to reach equilibrium were significantly different (Figure 3A,B). Compared to NF, the *RWA_m_* of cotton seed to NB decreased by 4.66%, and the time to reach constant increased by 8 h. Compared to MF, the *RWA_m_* of cotton seed to MB decreased by 2.22%, and the time to reach constant increased by 4 h. The *RWA_m_* of magnetized water was greater than non-magnetized water, while the time to reach equilibrium was lesser for magnetized water. Compared with NF, the *RWA_m_* of cotton seeds to MF increased by 16.76%, and the time to reach constant reduced by 3 h. Compared to NB, the *RWA_m_* of cotton seed to MB increased by 19.75%, and the time to reach equilibrium decreased by 7 h. Compared to NF, the *RWA_m_* of cotton seeds to MB increased by 14.17%, and the time to reach equilibrium increased by 1 h. 

The water absorption rate of magnetized and non-magnetized water decreased gradually with the increase in soaking time, and finally tended to zero. The water absorption curves of different water types were significantly different (Figure 3C,D). The maximum water absorption rate (*WAR_m_*) of cotton seeds to brackish water was lesser than fresh water. The *WAR_m_* of cotton seed to NB decreased by 36.25% as compared to NF, and the *WAR_m_* of cotton seed to MB decreased by 14.54% as compared to MF. The water absorption rate curves of the magnetized and non-magnetized water seeds had obvious intersection points. Before the intersection point was the magnetic effect time, the water absorption rate of seeds to magnetized water was greater than non-magnetized water. After the intersection point was the magnetic relaxation time, the absorption rate of seeds to magnetized water was close to or slightly lower than non-magnetized water. The *WAR_m_* of MF was 85.11% higher than NF, and the *WAR_m_* of MB was 148.14% higher than NB. The intersection point A_1_ of NF and MF was approximately 4 h after soaking, while the intersection point A_2_ of NB and MB was approximately 6 h after soaking. The intersection point of brackish water lagged behind fresh water by 2 h, and the magnetic effect time of brackish water was longer than fresh water. The *WAR_m_* of MB was greater than NF by 58.20%, and the *WAR_m_* of cotton seeds to brackish water after magnetization was higher than fresh water.

### 3.2. Germination Characteristics of Cotton Seed

Irrigation water quality and magnetization intensity had a noteworthy influence on the germination number of cotton seeds (*p* < 0.05, Figure 4). In the same period, the germination number of cotton seeds cultured in brackish water was lower than in fresh water, and germination number was higher in magnetized as compared to non-magnetized water. Due to the high salt content in brackish water, the germination number of cotton seeds in the same period decreased significantly. Two days after sowing, the germination number of cotton seeds cultivated with NB and MB decreased by 53.85% and 47.37%, compared with NF and MF, respectively. The germination number of cotton seeds cultivated with NB and MB decreased by 64.29% and 52.94%, respectively, compared with NF and MF after 2–4 days. The germination number of cotton seeds cultivated with NB and MB decreased by 33.33% and 30.00%, respectively, compared with NF and MF after 4–8 days. Eight days after sowing, as compared to NF and MF, the germination number of cotton seeds cultivated with NB and MB decreased by 55.28% and 43.18%, respectively. Magnetized water treatment could promote seed germination to a certain extent, and the germination number increased in the same period. Two days after sowing, the germination number of cotton seeds cultivated with MF and MB increased by 46.15% and 66.67%, respectively, compared with NF and NB. The germination number of cotton seeds cultivated with MF and MB increased by 21.43% and 60.00%, respectively, compared to NF and NB after 2–4 days. The germination number of cotton seeds cultivated with MF and MB increased by 11.11% and 16.67%, respectively, compared to NF and NB after 4–8 days. Eight days after sowing, the germination number of MF and MB cotton seeds increased by 22.22% and 47.06%, respectively, compared to NF and NB. Magnetized water had the greatest effect on the germination of cotton in the first four days after sowing, and the effect of magnetized brackish water on the germination of cotton seeds was greater than magnetized fresh water.

There were significant differences in the germination potential (*GP*), germination rate (*GR*), germination index (*GI*), and vigor index (*VI*) of cotton seeds under differing irrigation water quality and magnetized water treatment conditions (*p* < 0.05, Table 2). Brackish water could inhibit germination vigor and growth potential of seeds. Compared to NF, *GP*, *GR*, *GI,* and *VI* of seeds irrigated with NB decreased by 58.75%, 52.34%, 54.37%, and 73.49%, respectively. Compared to MF, *GP*, *GR*, *GI,* and *VI* of seeds irrigated with MB decreased by 47.52%, 41.98%, 43.02%, and 48.42%, respectively. Magnetized water treatment could enhance cotton seed vigor. Compared to NF, *GP*, *GR*, *GI,* and *VI* of seeds irrigated with MF improved by 26.25%, 22.43%, 26.95%, and 77.99%, respectively. Compared to NB, *GP*, *GR*, *GI,* and *VI* of cotton seeds irrigated with MB improved by 60.61%, 49.02%, 58.55%, and 246.33%, respectively. The impact of magnetized brackish water on the germination and growth potential of cotton seeds was greater than fresh water. 

### 3.3. Emergence Rate and Seedling Activity of Cotton

The emergence rate (*ER*) was found to improved gradually and then inclined to be stable with the increase in time for both magnetized and non-magnetized water, and the cotton seeds began to emerge on the third day (Figure 5). Compared to NF, the *ER* of cotton under NB irrigation decreased by 25.38%, and full seedling formation was delayed by 2 days. Compared to MF, the *ER* of cotton irrigated with MB decreased by 6.44%, and full seedling formation was delayed by 2 days. Magnetized water significantly enhanced the emergence rate of cotton (*p* < 0.05). Compared to NF, the *ER* of cotton irrigated with MF increased by 13.14%, and full seedling formation occurred one day earlier. Compared to NB, the *ER* of cotton irrigated with MB increased by 41.86%, and full seedling formation occurred one day earlier. Compared to NF, the *ER* of cotton irrigated with MB increased by 5.86%, and full seedling formation occurred one day earlier. The effect of magnetization of brackish water on the *ER* of cotton was greater than fresh water.

The seedling activity indexes of cotton including the seedling height (*Hs*), taproot length (*TLs*), fresh weight (*FWs*), dry weight (*DWs*), and moisture content (*MCs*) under brackish water irrigation were all significantly less than fresh water (*p* < 0.05, Table 3). Compared to the NF, the *Hs*, *TLs*, *FWs*, *DWs,* and *MCs* of seedlings irrigated with NB decreased by 44.69%, 36.23%, 55.23%, 33.97%, and 8.27%, respectively. Compared to MF, MB decreased the *Hs*, *TLs*, *FWs*, *DWs,* and *MCs* of seedlings by 20.49%, 23.60%, 51.39%, 26.79%, and 6.51%, respectively. Brackish water irrigation had the greatest effect on the *FWs* of seedlings. The seedling activity indexes of cotton irrigated with magnetized water improved significantly compared with non-magnetized water (*p* < 0.05). Compared to NF, MF improved the *Hs*, *TLs*, *FWs*, *DWs,* and *MCs* of seedlings by 7.96%, 28.99%, 66.42%, 22.52%, and 4.28%, respectively. Compared to NB, the *Hs*, *TLs*, *FWs*, *DWs,* and *MCs* of seedlings irrigated with MB increased by 55.20%, 54.55%, 76.36%, 35.84%, and 6.27%, respectively. It was thus clear that the effect of magnetized brackish water on the seedling activity indexes was significantly greater than magnetized fresh water. The *Hs*, *TLs*, *FWs*, *DWs,* and *MCs* of cotton seedlings under MB irrigation were all significantly higher than that under NF irrigation. These results illustrated that magnetized water reduced the inhibitory effect of brackish water on cotton seedlings and made brackish water reach or even exceed the efficiency of fresh water.

### 3.4. Morphological Development of Cotton Seedling

The morphological indexes of cotton seeding containing plant height (*Hp*), stem diameter (*SDp*), number of leaves (*LNp*), and single leaf area (*LAp*) were found to increase with growth time (Figure 6). In the same growth period, there were significant differences in the *Hp*, *SDp*, *LNp,* and *LAp* between fresh and brackish water (*p* < 0.05). Compared to NF and MF, the *Hp*, *SDp*, *LNp,* and *LAp* of cotton under NB and MB irrigation decreased by 36.48%, 39.21%, 40.00%, and 32.25%; 15.62%, 14.14%, 18.18%, and 15.53%, respectively, 40 d after germination. Compared to NF and NB, the *Hp*, *SDp*, *LNp,* and *LAp* of cotton irrigated with NB and MB increased by 51.07%, 47.35%, 46.67%, and 46.97%; 100.68%, 108.10%, 100.00%, and 83.24%, respectively, 40 days after final singling. Obviously, the impact of magnetized brackish water on seedling morphology indexes was greater than magnetized fresh water.

### 3.5. Morphological Development of Cotton Seedlings

The photosynthetic parameters of cotton seedlings irrigated with different magnetized water had significant differences (*p* < 0.05, Figure 7). Compared to NF, the net photosynthetic rate (*P_n_*), stomatal conductance (*G_s_*), intercellular CO_2_ concentration (*C_i_*), and transpiration rate (*T_r_*) of cotton irrigated with NB decreased by 16.25%, 19.64%, 25.85%, and 2.76%, respectively, while chlorophyll SPAD increased by 25.05%. Compared to MF, the *P_n_*, *G_s_*, *C_i_*, and *T_r_* of cotton under MB irrigation decreased by 9.23%, 30.53%, 24.11%, and 11.00%, respectively. Compared to NF, the *P_n_*, *G_s_*, *C_i_*, *T_r_*, and chlorophyll SPAD of MF irrigated cotton increased by 33.94%, 69.64%, 51.34%, 24.67%, and 91.10%, respectively. Compared with NB, the *P_n_*, *G_s_*, *C_i_*, *T_r_*, and chlorophyll SPAD of cotton irrigated with MB increased by 45.16%, 46.67%, 54.89%, 14.10%, and 27.06%, respectively. Hence, the increasing extent of *P_n_* of cotton under magnetized brackish water irrigation was larger than magnetized fresh water. 

Further analysis of seedling instantaneous water use efficiency (*iWUE*) and stomatal limit (*L_s_*) showed that there were significant differences in *iWUE* and *L_s_* of cotton among different qualities of magnetized water (*p* < 0.05, Table 4). Compared to NF, the *iWUE* of NB irrigated cotton decreased 14.58%, while *L_s_* increased by 23.39%. There was also a significant difference in *iWUE* and *L_s_* between MF and NF irrigated cotton (*p* < 0.05). Compared to NF, *iWUE* of MF irrigated cotton increased by 5.37%, while *L_s_* decreased by 4.65%. The difference in *iWUE* and *L_s_* between MB and NB irrigated cotton was significant (*p* < 0.05). Compared to NB, *iWUE* of MB irrigated cotton increased 25.14%, while *L_s_* reduced by 29.85%. From these, we could see that the increasing extent of *iWUE* of cotton irrigated with magnetized brackish water was larger than magnetized fresh water. 

### 3.6. Biomass and Its Allocation of Cotton Seedlings

The qualities of magnetized water had a significant influence on the seedling biomass and *its* allocation as shown in Table 5 (*p* < 0.05). Compared to NF and MF, the dry weight of stem (*DWs*), leaf (*DWl*), root (*DWr*), and total biomass (*DWt*) of cotton irrigated with NB and MB decreased by 12.96%, 14.10%, 32.91%, and 15.32%; 2.89%, 9.38%, 12.78%, and 7.80%, respectively, among which the *DWr* decreased the most. Compared to NF, the root-to-stem ratio of seedlings under NB irrigation decreased significantly (*p* < 0.05), with a range of 22.23%. The stem-to-total ratio and leaf-to-total ratio did not change significantly (*p* > 0.05). Compared to NF, the *DWs*, *DWl*, *DWr,* and *DWt* of seedlings irrigated with MF increased by 6.07%, 11.63%, 22.92%, and 10.87%, respectively. Compared to NB, the *DWs*, *DWl*, *DWr,* and *DWt* of cotton under MB irrigation increased by 18.35%, 17.76%, 59.81%, and 20.73%, respectively. The relative increase in cotton biomass in magnetized brackish water irrigation was greater than magnetized fresh water, and the relative increase in the *DWr* was the largest. Compared to NF, MF had no significant effect on the stem-to-total ratio, leaf-to-total ratio, and root shoot ratio of cotton (*p* > 0.05), while MB irrigation significantly increased the root-to-stem ratio as compared to cotton irrigated with NB by 35.72%.

### 3.7. Correlation Analysis of Physiological Growth Indexes

The physiological growth indicators of cotton seeds and seedlings were analyzed using Pearson correlation analysis under various magnetized water irrigation conditions (Figure 8). *RWA_m_* was found to have a significant positive connection with *Hp*, *SDp*, *LAp*, *P_n_*, and *DWs* (*p* < 0.05). At the same time, there was a substantial positive correlation between *WAR_m_* and *Hp*, *SDp*, *LNp*, *LAp*, *C_i_*, *T_r_*, and *DWs* (*p* < 0.05), and a very significant positive correlation between *WAR_m_* and *P_n_* (*p* < 0.01). It was discovered that by enhancing the water absorption ability of seeds, magnetized water might boost the photosynthetic rate of cotton seedlings and promote the accumulation of dry matter. The correlation coefficient between *GR* and *ER* was more than 0.9, showing that magnetized water could boost the emergence rate through enhancing cotton seed germination. *MCs* had a substantial positive association with *Hs*, *TLs*, and *FWs* (*p* < 0.05), as well as a very significant positive correlation with *DWs* (*p* < 0.01), implying that magnetized water could improve seedling vitality by increasing water content. *SDp*, *LNp*, and *P_n_* all had a significant positive correlation (*p* < 0.05), while *Hp*, *LAp*, and *P_n_* all had a very significant positive correlation (*p* < 0.01). It was shown that raising the photosynthetic rate of cotton using magnetized water could stimulate the development of aboveground morphology. Furthermore, *Ls* had a strong negative correlation with *TLs*, *Hp*, *SDp*, *LNp,* and *LAp*, indicating that magnetized water could stimulate cotton seedlings growth by lowering stomatal limitation of cotton leaves.

## 4. Discussion

### 4.1. Magnetic Water Treatment Improves Water Absorption, Germination, Emergence, and Seedling Vigor of Cotton Seeds

It is necessary for seeds to absorb enough water in the imbibition stage before germination [45]. Seed water absorption during imbibition depends on the physical water absorption of the protoplast colloid and is independent of seed metabolism [46,47]. The water absorption rate and relative water absorption of cotton seeds to brackish water were lower than fresh water (Figure 3), which may be due to the increase in solute potential by ions in brackish water, and the inhibition of seed water absorption [48]. Magnetized water treatment improved the relative water absorption capacity and water absorption rate of cotton seeds (Figure 3), which demonstrated that the water absorption of seeds was closely related to the physical and chemical properties of irrigation water. After the irrigation water was magnetized, the surface tension and contact angle of water decreased [15], and the pH, osmotic pressure, and solubility increased [49]. Meanwhile, many studies had demonstrated that the average distance between water molecules increased and some hydrogen bonds weakened or even broke in the microcosmic level of magnetized water [50]. These disassembled large water molecule clusters increased the amount of free dimer and monomer water molecules [51] and decreased the chemical bond angle and water ion gel radius [52]. Therefore, the changes in these physical and chemical properties of magnetized irrigation water could allow water to pass through the cell membrane easily, so as to promote imbibition and water absorption in cotton seeds and accumulate energy for seed germination. 

Seed germination involves the course of forming seedlings with normal roots, stems, and leaves under appropriate germination conditions after release of dormancy in active seeds [47]. The number of germinations, *GP*, *GR*, *GI,* and *VI* of cotton seeds irrigated with brackish water were lower than fresh water (Figure 4 and Table 2). This showed that salinity had a major inhibitory effect on the growth potential of seedlings, which was in agreement with previous studies by Mensah et al. (2006) [53], who pointed out that when electrical conductivity saline water was greater than 2.60 ms cm^−1^, salt significantly delayed the germination of peanut seeds and reduced the final germination rate. Additionally, magnetization of irrigation water could promote the germination of cotton seeds (Figure 4) and improve seed vigor (Table 2). Our findings on valuable effects of magnetized water are consistent also with the findings reported by Abedinpour et al. (2017) [54] for maize seeds. There are three key reasons for this phenomenon: (I) Magnetized water treatment can promote the formation of free radicals in the biofilm of testa and inner cells. High concentration of free radicals can increase the permeability of the cell membrane and the testa, so as to improve the transport of water and substances. (II) After magnetization, the activity and oxygen content of water molecules increases, which provides a beneficial water and oxygen environment that is conducive to seed germination. (III) Magnetized water treatment can promote the respiration of seed cells and improve the metabolism of seeds; thus, accelerating the germination of cotton seeds [20]. 

High emergence rate is crucial for high and stable yields, while the increase in irrigation water salinity can lead to low emergence rate, delayed emergence time, and irregular growth [55]. The results of this study showed that brackish water irrigation not only reduced the emergence rate of cotton seedlings, but also delayed the time of full seedling, compared to fresh water (Figure 5). It indicated that the salinity of brackish water would significantly reduce the emergence of cotton seedlings. Additionally, compared to fresh water, the vigor indexes such as *Hs*, *TLs*, *FWs*, *DWs,* and *MCs* decreased significantly under brackish water irrigation (Table 3). This validated a study by Krauss et al. (1999) [56], who pointed out that brackish water irrigation reduced root crown response of larch seedlings. The magnetized water irrigation significantly improved the emergence rate of cotton and reduced the time of full seedling emergence (Figure 5). These findings on available effects of magnetized water are in agreement with the findings reported by Moussa (2011) [57] for snow pea and chickpea. Moreover, the increase in emergence rate and the decrease in time required for full seedling emergence were greater after magnetized brackish water irrigation compared to magnetized fresh water (Figure 5). This may be due to the strengthening of magnetization by mineral ions in brackish water [58]. Our study also pointed out that after magnetization of fresh and brackish water, the activity indexes of cotton seedlings increased (Table 3), which indicated that magnetization treatment could mitigate the inhibitory effect of salt on crop vigor to a certain extent and enhance the emergence rate of crops [59].

### 4.2. Magnetic Water Treatment Promotes Development of Cotton Root System and Morphology, and Changes the Distribution Pattern of Biomass

The root architecture has a direct impact on crop growth and yield [55]. The growth of the root system of cotton seedlings was significantly inhibited by brackish water irrigation. Taproot length (*TLs*, Table 3) and root weight (*DWr*, Table 5) were significantly reduced, which inevitably reduced the intensity and range of the root system to absorb nutrients and water [29]. The *TLs* (Table 3) and *DWr* (Table 5) of cotton seedlings under magnetized water irrigation was considerably higher than non-magnetized water, which revealed that magnetic water treatment could boost root growth, enhance the selective absorption capacity for mineral ions, and prevent excessive uptake of Na^+^ by cells; thus, alleviate the inhibitory impact of salt stress on growth of cotton seedlings [44]. It has been reported that the young roots of crops show a magnetic orientation in magnetic fields [60]. Magnetized water can improve absorption of nutrients by roots by stimulating root tissue differentiation for the root elongation volume increasing and radicle elongation [55]. Consequently, magnetic water treatment can improve absorption of water and nutrition in crops and ameliorate the economic properties of crops during later stages [23].

The increase in salinity of irrigation water can inhibit the morphological development and biomass accumulation of cotton [58]. Compared to fresh water, the *DWs*, *DWl*, *DWr* (Table 5), and morphological indexes such as *Hp*, *SDp*, *LNp,* and *LAp* (Figure 6) of cotton irrigated with brackish water decreased significantly, which indicated that the salinity of brackish water would inhibit the morphological development and biomass accumulation of cotton seedlings [47]. Compared to non-magnetized water, the *DWs*, *DWl*, *DWr* (Table 5), and morphological indexes of the *Hp*, *SDp*, *LNp,* and *LAp* (Figure 6) of cotton seedlings irrigated with magnetic water increased significantly, which confirmed a recent study reported by Abedinpour et al. (2017) [54]. The relative increase in morphological indexes of cotton seedlings under magnetized brackish water treatment was larger than magnetized fresh water. The influence of magnetized fresh water on the stem-to-total ratio, leaf-to-total ratio, and root-to-stem ratio of cotton seedlings was lesser, while the impact of magnetized brackish water on the root-to-stem ratio of cotton seedlings was significantly higher, which revealed that the influence of magnetic treatment water on cotton seedlings growth was closely related to the irrigation water quality [17]. Magnetized brackish water can promote the accumulation of total biomass of cotton seedlings by improving the dry matter ratio of roots to absorb more water and nutrients [29]. 

### 4.3. Magnetic Water Treatment Improves Utilization Efficiency of Light Energy and Water in Cotton

Increased salinity of irrigation water can damage photosynthetic organs, inhibit or damage the photosynthetic electron transport system, and reduce photosynthetic intensity of crops [44]. This study showed that brackish water irrigation resulted in the decrease in *P_n_*, *G_s_*, *C_i_*, *T_r_*, and *WUE*, while the increasing chlorophyll SPAD value and *L_s_* (Figure 7 and Table 4). This indicated that the increase in the salinity of irrigation water could improve chlorophyll content of cotton leaves to a certain extent, but it would lead to the increase in stomatal limitation, and then the decrease in light energy and water use efficiency of cotton. This is primarily because brackish water contains a large number of trace elements, which is conducive to the formation of chlorophyll [58]. Simultaneously, brackish water also contains a lot of Na^+^ (Table 1), excessive accumulation of Na^+^ causes damage to cell membrane system, a decrease in cell osmotic potential, and an increase in stomatal restriction; thus, decreasing photosynthetic rate and water use efficiency [61]. 

We found that the cotton seedlings had higher *P_n_*, *G_s_*, *C_i_*, *T_r_*, *WUE*, chlorophyll SPAD values, and lower *L_s_* values under magnetized water irrigation than non-magnetized water (Figure 7 and Table 4). This illustrates that magnetic water treatment can noteworthily stimulate the stomatal conductance of cotton seedlings, alleviate stomatal limitation, and accelerate the supply of CO_2_. Ultimately, the utilization efficiency of light energy and water in cotton is improved owing to the enhanced capacity of photosynthetic carbon assimilation [62]. Further, the increase in light and water use efficiency of magnetized brackish water irrigated cotton was greater than magnetized fresh water. This demonstrates that the improvement in light use efficiency by magnetized water treatment of cotton is closely related to irrigation water quality. It also indicates that the magnetized brackish water is more beneficial than magnetized fresh water, which may be because the more mineral ions in brackish water can strengthen the magnetization force and improve the physical and chemical properties [63]. In previous studies, we found that magnetized water improved the metabolism rate of crops by stimulating the activity of related enzymes [64]. Furthermore, the increasing photochemical activity of chlorophyll and the amount of free water in magnetized crop cells could contribute to the increasing net photosynthetic rate [65]. In general, our results of photosynthesis further confirm the biomagnetic properties of magnetic water technology in crops.

## 5. Conclusions

By comparing the effects of different qualities (fresh and brackish water) of magnetized water on seed water absorption, germination, and seedling growth of cotton, it was found that fresh water was more easily absorbed by cotton seeds than brackish water, while magnetized water was more easily absorbed by cotton seeds than non-magnetized water. Magnetized water could promote seed germination and germination vigor by increasing seed water absorption. The relative promotion effect of magnetized brackish water on cotton seed germination vigor and growth potential was greater than magnetized fresh water. In addition, magnetized water could increase the emergence rate and promote the morphology development of cotton. When compared to unmagnetized magnetized brackish water and unmagnetized fresh water, magnetic brackish water had a stronger relative promoting effect on cotton growth than magnetized fresh water. Furthermore, magnetized water could promote the utilization efficiency of light energy and water of cotton seedlings and increase cotton biomass. The effect of magnetic treatment fresh water on stem-to-total ratio, leaf-to-total ratio, and root-to-stem ratio of cotton seedlings was not significant, but root-to-stem ratio of cotton irrigated with magnetized brackish water was increased by 35.72%. Therefore, it is suggested that magnetized brackish water can be used to irrigate cotton seedlings when freshwater resources are insufficient. Although the effects of magnetized water qualities on early growth of cotton was shown above, we cannot ignore the magnetization intensity effect on the growth and yield of cotton. The combination of qualities and magnetization intensity may further improve the yield and quality of cotton in the late growth stage, but more research is needed in the near future.

## Figures and Tables

**Figure 1 plants-11-01397-f001:**
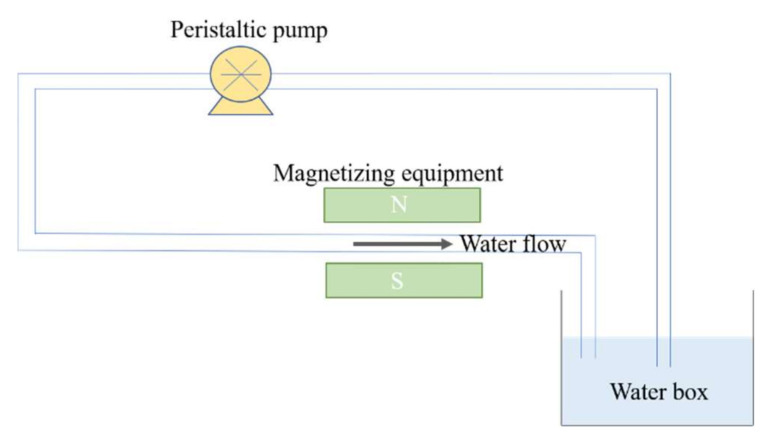
Schematic diagram of the magnetized water device.

**Figure 2 plants-11-01397-f002:**
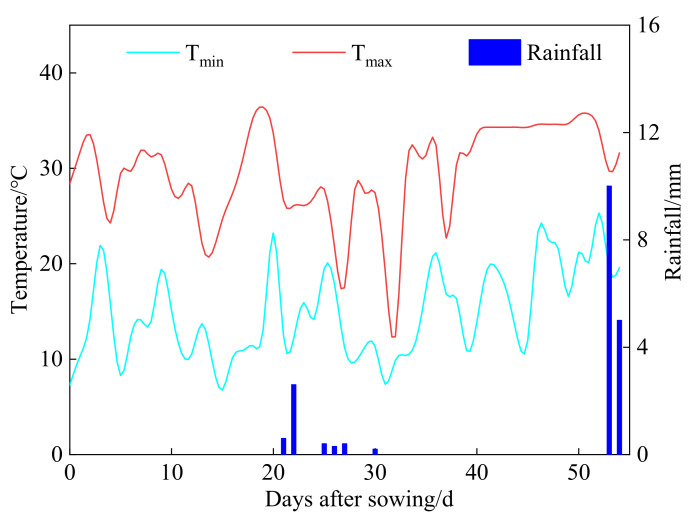
The temperature and rainfall during the field pot experiment in 2018. T_min_ and T_max_ represents the minimum and maximum temperatures, respectively.

**Figure 3 plants-11-01397-f003:**
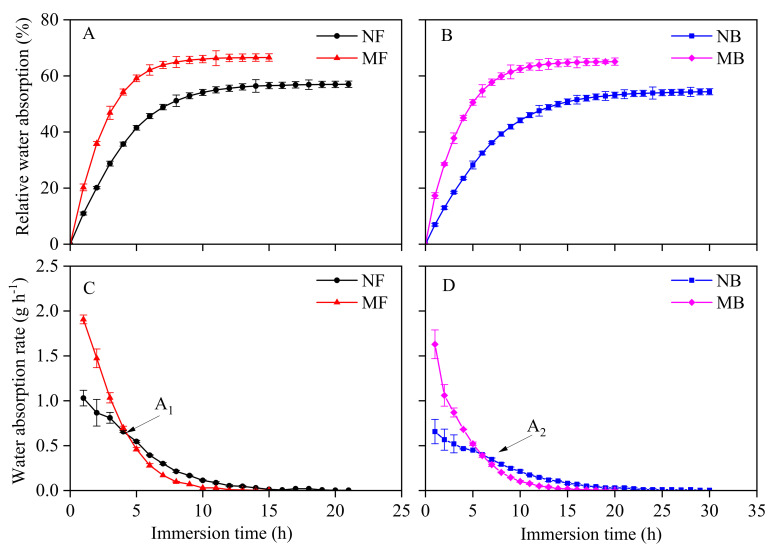
Changes in relative water absorption and water absorption rate (mean  ±  S.D. for three replicates) of cotton seeds. (**A**) Changes in relative water absorption NF and MF. (**B**) Changes in relative water absorption NB and MB. (**C**) Changes in water absorption rate NF and MF. (**D**) Changes in water absorption rate NB and MB. NF represents non-magnetized fresh water, MF represents magnetized fresh water, NB represents non-magnetized brackish water, and MB represents magnetized brackish water. The same below. A_1_ is the intersection point of the water absorption rate curve of NF and MF, and A_2_ is the intersection point of the water absorption rate curve of NB and MB.

**Figure 4 plants-11-01397-f004:**
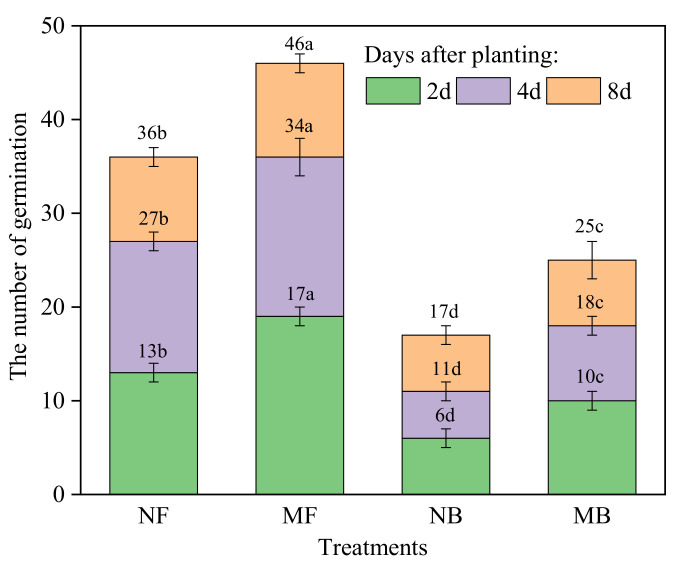
Effect of different qualities of magnetized water irrigation on germination number of cotton seeds. Different lowercase letters indicate significant difference at *p* < 0.05.

**Figure 5 plants-11-01397-f005:**
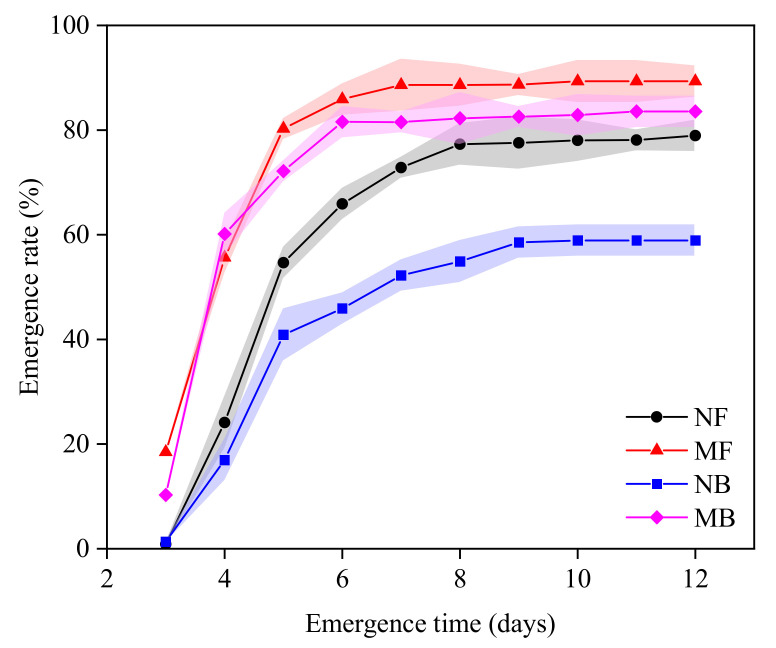
Curve of cotton emergence rate under different qualities of magnetized water irrigation.

**Figure 6 plants-11-01397-f006:**
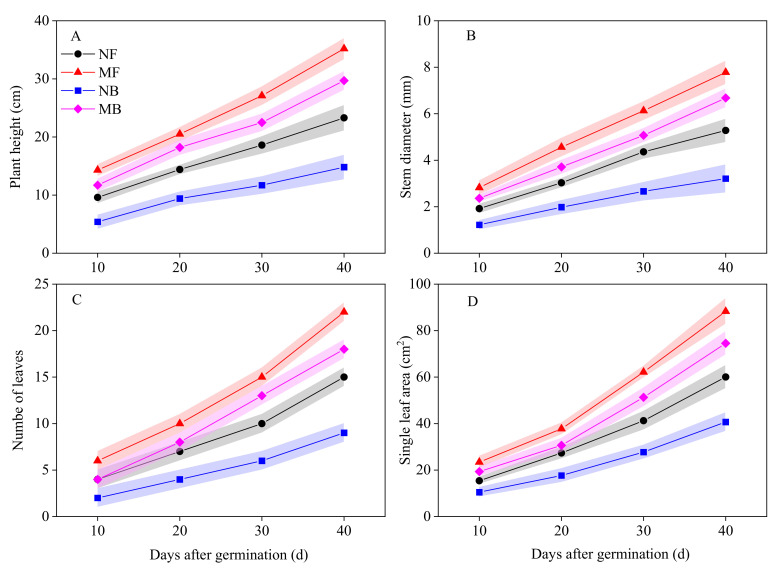
Effect of different magnetized water irrigation on seeding morphological indexes of cotton. (**A**) Plant height, (**B**) Stem diameter, (**C**) Number of leaves, (**D**) Single leaf area.

**Figure 7 plants-11-01397-f007:**
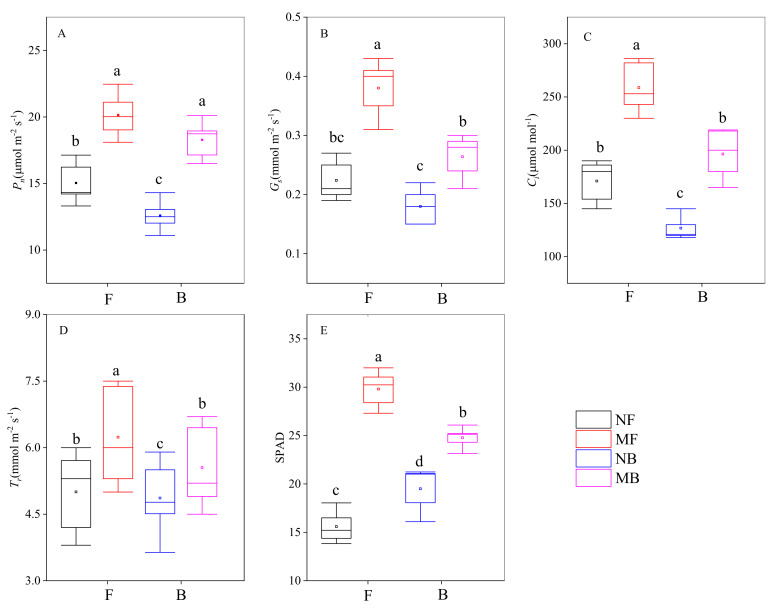
Effect of different magnetized water irrigation on photosynthetic parameters of cotton. (**A**) *P_n_*, (**B**) *G_s_*, (**C**) *C_i_,* (**D**) *T_r_*, (**E**) SPAD. The box boundaries indicate the 25th and 75th percentiles, the lines in the boxes mark the median, and lines below and above the boxes indicate the 10th and 90th percentiles, respectively. F and B represent fresh water and brackish water, respectively. *P_n_* represents the net photosynthetic rate, *G_s_* represents the stomatal conductance, *C_i_* represents the intercellular CO_2_ concentration, and *T_r_* represents the transpiration rate. Different lowercase letters indicate significant difference at *p* < 0.05.

**Figure 8 plants-11-01397-f008:**
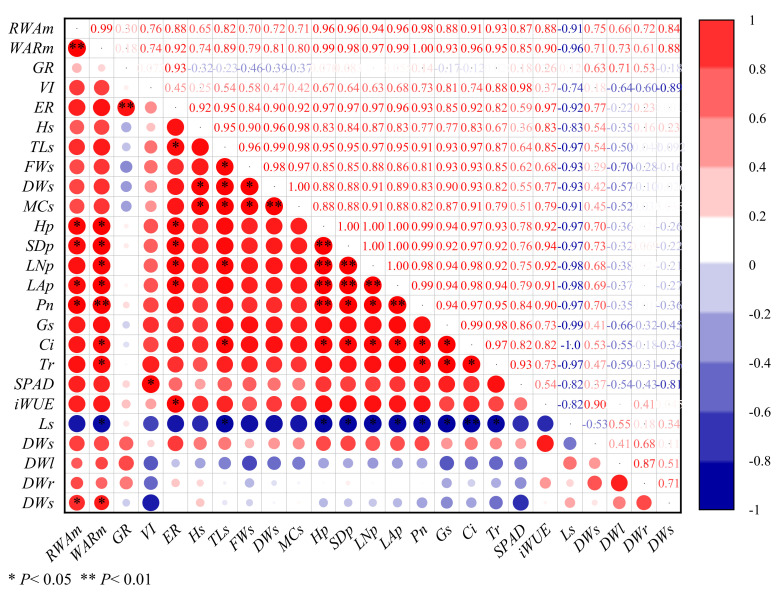
Correlation analysis of physiological growth indicators of cotton seeds and seedlings irrigated with magnetized water.

**Table 1 plants-11-01397-t001:** Chemical composition of the fresh and brackish water.

Water Quality Type	Salinity (g L^−1^)	pH	Anions (mmol L^−1^)				Cations (mmol L^−1^)				SAR
			CO_3_^2−^	HCO_3_^−^	Cl^−^	SO_4_^2−^	Ca^2+^	Mg^2+^	Na^+^	K^+^	(mmol L^−1^)^0.5^
Fresh water	0.72	7.80	0.12	4.91	2.21	1.66	1.82	1.61	2.37	0.09	1.81
Brackish water	2.75	7.84	6.18	5.87	12.58	11.94	4.68	6.58	17.73	0.57	5.29

**Table 2 plants-11-01397-t002:** Seeds germination indexes of cotton under different qualities of magnetized water irrigation.

Treatments	*GP* (%)	*GR* (%)	*GI*	*VI*
NF	53.3 ± 1.2 b	71.3 ± 1.2 b	17.6 ± 0.7 b	26.4 ± 0.8 b
MF	67.3 ± 3.1 a	87.3 ± 1.2 a	22.4 ± 0.8 a	47.0 ± 2.5 a
NB	22.0 ± 2.0 d	34.0 ± 2.0 d	8.0 ± 0.3 d	7.0 ± 1.4 d
MB	35.3 ± 1.2 c	50.7 ± 4.2 c	12.8 ± 0.5 c	24.2 ± 1.7 c

*GP*, *GR*, *GI,* and *VI* represent the germination potential, germination rate, germination index, and vigor index of cotton seeds, respectively. Different lowercase letters indicate significant difference at *p* < 0.05.

**Table 3 plants-11-01397-t003:** Seedling activity indexes of cotton under different qualities of magnetized water irrigation.

Treatments	*Hs* (cm)	*TLs* (cm)	*FWs* (g)	*DWs* (g)	*MCs* (%)
NF	7.5 ± 0.1 b	4.6 ± 0.2 b	5.89 ± 0.11 b	0.87 ± 0.02 b	85.17 ± 0.04 b
MF	8.1 ± 0.1 a	5.9 ± 0.2 a	9.57 ± 0.28 a	1.07 ± 0.03 a	88.81 ± 0.06 a
NB	4.2 ± 0.1 d	2.9 ± 0.2 d	2.64 ± 0.07 c	0.58 ± 0.02 d	78.13 ± 0.09 d
MB	6.5 ± 0.1 c	4.5 ± 0.4 c	4.65 ± 0.42 b	0.78 ± 0.03 c	83.03 ± 2.09 c

*Hs*, *TLs*, *FWs*, *DWs,* and *MCs* represent the seedling height, taproot length, fresh weight, dry weight, and moisture content, respectively. Different lowercase letters indicate significant difference at *p* < 0.05.

**Table 4 plants-11-01397-t004:** The *iWUE* and *L_s_* of cotton seedlings irrigated with different qualities of magnetized water.

Treatments	*iWUE* (μmol mmol^−1^)	*L_s_*
NF	3.14 ± 0.89 b	0.53 ± 0.06 b
MF	3.30 ± 0.58 a	0.28 ± 0.07 c
NB	2.68 ± 0.63 c	0.65 ± 0.03 a
MB	3.35 ± 0.46 a	0.45 ± 0.07 b

*iWUE* and *L_s_* represent the instantaneous water use efficiency and stomatal limit of cotton seedlings, respectively. Different lowercase letters indicate significant difference at *p* < 0.05.

**Table 5 plants-11-01397-t005:** Biomass and its allocation of cotton seedlings irrigated with different qualities of magnetized water.

Treatments	*DWs* (g)	*DWl* (g)	*DWr* (g)	*DWt* (g)	Stem-to-Total Ratio (%)	Leaf-to-Total Ratio (%)	Root-to-Stem Ratio (%)
NF	5.76 ± 0.12 b	11.49 ± 0.2 b	1.57 ± 0.12 b	18.82 ± 0.42 b	30.61 ± 0.21 ab	61.06 ± 0.33 ab	9.10 ± 0.57 a
MF	6.11 ± 0.12 a	12.83 ± 0.15 a	1.93 ± 0.09 a	20.87 ± 0.18 a	29.28 ± 0.33 b	61.47 ± 0.18 ab	10.2 ± 0.62 a
NB	5.01 ± 0.07 c	9.87 ± 0.19 c	1.05 ± 0.09 c	15.94 ± 0.22 c	31.47 ± 0.85 a	61.93 ± 0.35 a	7.07 ± 0.58 b
MB	5.93 ± 0.17 ab	11.62 ± 0.22 b	1.68 ± 0.13 b	19.24 ± 0.25 b	30.84 ± 0.51 ab	60.41 ± 0.36 ab	9.60 ± 0.92 a

*DWs*, *DWl*, *DWr*, and *DWt* represent the dry weight of stem, leaf, root, and total biomass of cotton seedlings, respectively. Different lowercase letters indicate significant difference at *p* < 0.05.

## Data Availability

The data presented in this study are available on request from the corresponding author. The data are not publicly available due to the project not being completed.

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
