# Peer review of "Magnetic Water Treatment: An Eco-Friendly Irrigation Alternative to Alleviate Salt Stress of Brackish Water in Seed Germination and Early Seedling Growth of Cotton (Gossypium hirsutum L.)"

_plants, 2022, doi:10.3390/plants11111397_

Round 1
Reviewer 1 Report
Comments and suggestions for Authors
Article entitled: ”Magnetic water treatment: An eco-friendly irrigation alternative to alleviate salt stress of brackish water in seed germination and early seedling growth of cotton (Gossypium hirsutum L.)”
The topic is very interesting and falls within the scope of the journal. The experimental dataset is useful and scientifically valuable. The presented research concerns the current global problem. The aim of the study was different influences of magnetized fresh and brackish water on early growth of cotton were compared and analyzed via seed water absorption characteristics, germination parameters, seedling growth, biomass, photosynthesis parameters and chlorophyll content. However, the one-year test cycle may be a problem in correctly concluding.
General remarks
In order to increase the usefulness of the article, Authors must refer to the following points.
Additions should be made to increase the scientific value of the manuscript.
- Abstract - There are no years of doing research.
- Introduction - A research hypothesis is missing.
- Materials and Methods – Section 2.1. - It is necessary to supplement the distribution of rainfall and temperature during the pot experiment in the field. Section 2.3.3. - You need to complete the soil type, soil granulometric composition in pots and pH. The name of the statistical method according to which the pot experiment was carried out must be completed. The number of research factors should be given. You must enter the number of pots in the experiment. Complete the date of sowing cotton seeds.
- Results - are well described and clearly presented in Tables and Figures. Only Figure 7 requires the readability of the values of correlation coefficients to be improved.
- Discussion – The discussion is very well written and organized.
- The conclusions are good. I have no comments.
Specific comments
Line 323 - should be Table 3…...
Figure 5C – should be: Number of leaves
The References must be corrected according to the editorial requirements of the publisher (bold-years, DOI, etc).
Author Response
- Abstract - There are no years of doing research.
Response: The year of doing research has been added in the Abstract.
- Introduction - A research hypothesis is missing.
Response: The research hypothesis has been added in the Introduction. “The hypothesis to be studied is whether the quality of magnetized water affects cotton seed water absorption, germination, and seedling growth in different ways.”
- Materials and Methods – Section 2.1. - It is necessary to supplement the distribution of rainfall and temperature during the pot experiment in the field. Section 2.3.3. - You need to complete the soil type, soil granulometric composition in pots and pH. The name of the statistical method according to which the pot experiment was carried out must be completed. The number of research factors should be given. You must enter the number of pots in the experiment. Complete the date of sowing cotton seeds.
Response: The temperature and rainfall during the field pot experiment have been added in Section 2.1.
The soil type, soil granulometric composition in pots, and pH have been added in Section 2.3.3. “The air-dried field soil from the upper 20 cm of the cotton field was used, which was sandy loam, according to the USDA's soil texture classification ( 64.27% sand, 32.83 % silt, 2.90 % clay ). The total N, available P, available K, ECe (electrical conductivity of soil saturation extract) and pH were 23.81 mg kg-1, 15.23 mg kg-1, 107.56 mg kg-1, 5.47 ds m-1, and 7.8, respectively [43].”
The name of the statistical method according to which the pot experiment was carried out has been completed. “Random pot trials were conducted on cotton fields, and the pot experiment was statis-tically assessed using analysis of variance.”
The number of research factors and pots in the experiment has been added. “Random pot trials were conducted on cotton fields, and the pot experiment was statis-tically assessed using analysis of variance. With a total of 12 pots, four treatments were set up based on varied water quality (NF, MF, NB,and MB), and each treatment was performed three times.”
The date of sowing cotton seeds has been completed. “On April 22, 2018, cotton seeds were sown.”
- Results - are well described and clearly presented in Tables and Figures. Only Figure 7 requires the readability of the values of correlation coefficients to be improved.
Response: The font of the correlation coefficient in Figure 7 has been increased to improve the readability, as suggested by the reviewer.
- Line 323 - should be Table 3…...
Response: It has been altered.
- Figure 5C – should be: Number of leaves
Response: It has been revised.
- The References must be corrected according to the editorial requirements of the publisher (bold-years, DOI, etc).
Response: The References are double-checked to meet the editorial requirements of the publisher.

Reviewer 2 Report
In this article, the authors analyzed the effects of different types of magnetized water with varying quality on seed absorption, germination and early growth of cotton. They showed that the biological effect of magnetized brackish water was more pronounced. These results have practical implications: magnetized brackish water can be used to irrigate cotton seedlings when fresh water resources are insufficient.
Specific comments:
Abstract (lines 20-22) states «The cotton seeds germination rate 20 under magnetized fresh and brackish water irrigation increased by 13.14% and 41.86%, respectively». However, Fig 3 shows that seed germination rates are greatly reduced in higher salinity conditions, and even the use of magnetized water cannot restore them to control levels.
Obtained results demonstrate that magnetization of brackish water does improve growth characteristics of cotton. This is an important result, but it is not clearly announced or suitably emphasized in the abstract and summary.
Introduction.
The authors conduct their studies using water with increased salinity to demonstrate that in such conditions, which are likely reflective of the conditions that are common in the region of interest to the authors, water magnetization has a beneficial effect on cotton germination and growth. However, the impact of salinity on the conclusions of the study is not sufficiently emphasized. It is important to discuss cotton growth properties in different conditions, and stress the role of salinity levels on study results.
Materials and methods. It is unclear, why the following units (m∙s-1) were chosen to express the water flow rate instead of liters per minute (L∙min-1). The volume per minute of pumped fresh and brackish water is important (line 122)
Discussion. The authors’ comparison of the action of magnetized water with the function of phytohormone is not supported by the data. Firstly, some phytohormones actively decrease the growth and development of plants. Secondly, the action of magnetized water on the molecular level is completely detached from the molecular mechanisms through which phytohormones act (lines 449-450). The authors need to discuss plausible mechanisms of action for magnetized brackish water, rather than simply stating that magnetization has a beneficial effect.
Conclusion.
The statement «The relative promoting effect of magnetic treatment brackish water on cotton growth was greater than magnetized fresh water» is not supported by the data, since Table 5 shows that certain growth characteristics are not significantly improved when using brackish vs fresh water. It is important to accurately summarize the selective effect of such treatment.
The conclusions are very clear, but it is also necessary to present some physiological reasoning for the action of magnetism.
Author Response
- Abstract (lines 20-22) states «The cotton seeds germination rate 20 under magnetized fresh and brackish water irrigation increased by 13.14% and 41.86%, respectively». However, Fig 3 shows that seed germination rates are greatly reduced in higher salinity conditions, and even the use of magnetized water cannot restore them to control levels.
Obtained results demonstrate that magnetization of brackish water does improve growth characteristics of cotton. This is an important result, but it is not clearly announced or suitably emphasized in the abstract and summary.
Response: The corresponding parts have been amended in accordance with the reviewers' comments. “The cotton seeds germination rate under magnetized fresh and magnetized brackish water irriga-tion relatively increased by 13.14% and 41.86%, respectively, and the relative promoting effect of magnetized brackish water on the vitality indexes and the morphological indexes of cotton seedlings was more significant than magnetized fresh water.” “Therefore, the magnetization of brackish water does improve the growth characteristics of cotton seedlings, and the biological effect of magnetized brackish water is more significant than that of fresh water.”
- Introduction. The authors conduct their studies using water with increased salinity to demonstrate that in such conditions, which are likely reflective of the conditions that are common in the region of interest to the authors, water magnetization has a beneficial effect on cotton germination and growth. However, the impact of salinity on the conclusions of the study is not sufficiently emphasized. It is important to discuss cotton growth properties in different conditions, and stress the role of salinity levels on study results.
Response: The impact of salinity levels of irrigation water on crops has been emphasized in the Introduction in accordance with the reviewers' comments. “It should be noted that when the salinity of brackish water surpasses 3 g/L, plant physiological growth exhibits salt stress symptoms [8].”
- Materials and methods. It is unclear, why the following units (m∙s-1) were chosen to express the water flow rate instead of liters per minute (L∙min-1). The volume per minute of pumped fresh and brackish water is important (line 122)
Response: According to the reviewer's comments, the flow rate unit has been converted to L min-1 in the Materials and methods.
- Discussion. The authors’ comparison of the action of magnetized water with the function of phytohormone is not supported by the data. Firstly, some phytohormones actively decrease the growth and development of plants. Secondly, the action of magnetized water on the molecular level is completely detached from the molecular mechanisms through which phytohormones act (lines 449-450). The authors need to discuss plausible mechanisms of action for magnetized brackish water, rather than simply stating that magnetization has a beneficial effect.
Response: The section on plant hormones has been removed in the Discussion, according to the reviewer's comments.
- Conclusion. The statement «The relative promoting effect of magnetic treatment brackish water on cotton growth was greater than magnetized fresh water» is not supported by the data, since Table 5 shows that certain growth characteristics are not significantly improved when using brackish vs fresh water. It is important to accurately summarize the selective effect of such treatment.
The conclusions are very clear, but it is also necessary to present some physiological reasoning for the action of magnetism.
Response: The corresponding parts have been amended in accordance with the reviewers' comments. “When compared to unmagnetized magnetized brackish water and unmagnetized fresh water, magnetic brackish water had a stronger relative promoting effect on cotton growth than magnetized fresh water.”
